# Different Strategies of Preimplantation Genetic Testing for Aneuploidies in Women of Advanced Maternal Age: A Systematic Review and Meta-Analysis

**DOI:** 10.3390/jcm10173895

**Published:** 2021-08-30

**Authors:** Wei-Hui Shi, Zi-Ru Jiang, Zhi-Yang Zhou, Mu-Jin Ye, Ning-Xin Qin, He-Feng Huang, Song-Chang Chen, Chen-Ming Xu

**Affiliations:** 1International Peace Maternity and Child Health Hospital, School of Medicine, Shanghai Jiao Tong University, 910 Hengshan Road, Shanghai 200030, China; Ashely_swh@sjtu.edu.cn (W.-H.S.); zzyoung@sjtu.edu.cn (Z.-Y.Z.); yemujin@sjtu.edu.cn (M.-J.Y.); huanghefg@sjtu.edu.cn (H.-F.H.); 2Shanghai Key Laboratory of Embryo Original Diseases, 145 Guangyuan Road, Shanghai 200030, China; 3Obstetrics and Gynecology Hospital, Fudan University, 566 Fangxie Road, Shanghai 200011, China; jiangziru@sjtu.edu.cn; 4Department of Assisted Reproductive Medicine, Shanghai First Maternity and Infant Hospital, Tongji University School of Medicine, Shanghai 200030, China; qinningxin@sjtu.edu.cn

**Keywords:** preimplantation genetic testing for aneuploid, advanced maternal age, comprehensive chromosomal screening, embryo biopsy

## Abstract

Background: Preimplantation genetic testing for aneuploidies (PGT-A) is widely used in women of advanced maternal age (AMA). However, the effectiveness remains controversial. Method: We conducted a comprehensive literature review comparing outcomes of IVF with or without PGT-A in women of AMA in PubMed, Embase, and the Cochrane Central Register of Controlled Trials in January 2021. All included trials met the criteria that constituted a randomized controlled trial for PGT-A involving women of AMA (≥35 years). Reviews, conference abstracts, and observational studies were excluded. The primary outcome was the live birth rate in included random control trials (RCTs). Results: Nine randomized controlled trials met our inclusion criteria. For techniques of genetic analysis, three trials (270 events) performed with comprehensive chromosomal screening showed that the live birth rate was significantly higher in the women randomized to IVF/ICSI with PGT-A (RR = 1.30, 95% CI 1.03–1.65), which was not observed in six trials used with FISH as well as all nine trials. For different stages of embryo biopsy, only the subgroup of blastocyst biopsy showed a higher live birth rate in women with PGT-A (RR = 1.36, 95% CI 1.04–1.79). Conclusion: The application of comprehensive chromosome screening showed a beneficial effect of PGT-A in women of AMA compared with FISH. Moreover, blastocyst biopsy seemed to be associated with a better outcome than polar body biopsy and cleavage-stage biopsy.

## 1. Introduction

Preimplantation genetic testing (PGT) was introduced over thirty years ago as an early form of prenatal genetic diagnosis performed with multiple assisted reproductive technologies, such as embryo biopsy, embryo vitrification, and embryo transfer [1]. This procedure is mostly performed to reduce the transmission of genetic disorders for patients with monogenic diseases (PGT-M), chromosomal segmental rearrangement (PGT-SR), or aneuploidies (PGT-A) [2]. It is well known that the karyotypically embryotic anomaly is one of the main causes of recurrent pregnancy losses [3]. The occurrence of embryotic aneuploidies increases with maternal age. As a result, women of advanced maternal age (AMA) have a higher risk of miscarriages, implantation failures after in vitro fertilization (IVF), fetal malformations, and fetuses born with chromosomal disorders [4,5]. However, the effectiveness of PGT-A in women of AMA is still controversial.

A meta-analysis reported in 2011 showed no evidence of a beneficial effect of PGT-A on live birth rate after IVF [6]. Possible reasons for the disappointing results might be associated with the PGT-A procedure per se, such as damages caused by embryo biopsy and misdiagnoses owing to embryotic mosaicism. As reported, blastomere biopsy might delay the compaction and blastulation of embryos, thus affecting the implantation [7,8]. Additionally, fluorescence in situ hybridization (FISH), the first and the most widely applied in PGT-A, can only assess a select number of chromosomes (5 to 12 chromosomes) simultaneously [9]. Furthermore, the accuracy of FISH was always affected by signal overlaps, signal splits, signal diffusion, and probe inefficiency [10,11].

In the past decade, PGT has developed rapidly with advanced methods of biopsy and strategies of genetic analysis. On the one hand, embryo biopsy has evolved into trophectoderm biopsy, which entails removing 5–10 trophectoderm cells at the blastocyst stage, which reported more accuracy than cleavage-stage biopsy and has less impact on embryo viability, leading to a higher implantation rate [12,13,14]. On the other hand, techniques of genetic testing have been developed from FISH to the comprehensive chromosome screening (CCS) technology, including real-time quantitative PCR (qPCR), array comparative genomic hybridization (aCGH), and next-generation sequencing (NGS) [15,16,17,18]. Although the qPCR method is ineffective in the detection of segmental aneusomy, an improvement in implantation and delivery rates has been observed in qPCR-based PGT-A [19]. Better than the qPCR method, aCGH has been commonly used in detection of segmental abnormalities and unbalanced translocations in PGT procedures [20]. Moreover, NGS is introduced as a meaningful approach of aneuploidy screening characterized by its high throughput, low cost, and high sensitivity and specificity. NGS can also be used in the detection of copy number variations, unbalanced translocations, and point mutations [21].

With the development of advanced technologies, the efficacy of PGT-A in women of AMA requires further investigation. Here, we conducted a systematic review of published RCTs to evaluate whether the PGT-A procedure is beneficial for pregnancy outcomes in women of AMA with IVF.

## 2. Materials and Methods

### 2.1. Search Strategy

We searched PubMed, Embase, and the Cochrane Central Register of Controlled Trials (CENTRAL) in January 2021 using the following search criteria: (preimplantation genetic diagnosis OR PGD OR preimplantation genetic screening OR PGS OR preimplantation genetic testing for aneuploidy OR PGT-A OR embryo screening OR preimplantation testing OR screening for aneuploidies) AND (randomized trial OR randomized controlled trial OR randomized study OR randomized). Additionally, we carefully reviewed the reference lists of included studies for relevant studies.

### 2.2. Study Selection

The result of the electronic search was independently examined by the two authors (Z.-R.J. and W.-H.S.) to select relevant trials. After removing duplicates, studies were included based on the criteria that compared the IVF/ICSI outcomes in women of AMA (over 35 years of age) with PGT-A or without PGT-A. If a trial involved women of different age groups, the data regarding the women of AMA were extracted. All included trials met the criteria that constituted a randomized controlled trial for PGT-A involving women of AMA. The trials that had mixed outcomes of women under 35 years of age with those of AMA were excluded. Reviews, conference abstracts, and observational studies were also excluded. Additionally, studies with the indication of recurrent miscarriage, repeated implantation failure, severe male infertility, or good-prognosis patients were excluded.

### 2.3. Risk of Bias Assessment

Included studies were assessed by two review authors (Z.-R.J. and W.-H.S.) using the Cochrane Collaboration’s tool [22]. Briefly, the tool covers six domains of bias assessments: selection bias, performance bias, detection bias, attrition bias, reporting bias, and other bias. The risk of bias graph and the risk of bias summary were constructed with Review Manager software (Revman Version 5.4, The Cochrane Collaboration, London, UK).

### 2.4. Statistical Analysis

The effect of PGT-A was assessed in different techniques of genetic testing and different stages of embryo biopsy. We calculated the risk ratio (RR) with 95% confidence intervals (CIs) for the occurrence of outcome events, including live birth rate as the primary outcome and ongoing pregnancy rate, clinical pregnancy rate, multiple pregnancy rate, and miscarriage rate as the secondary outcome measures. Statistical heterogeneity across studies was evaluated by the I2 statistic, which was considered to be positive with a value of ≥50%. For heterogeneity among several trials, the random effects model was used to pool the data for each subgroup separately and for all included studies. The meta-analysis was conducted using Review Manager software (Revman Version 5.4, The Cochrane Collaboration, London, UK).

## 3. Results

### 3.1. Results of the Search

The literature search identified a total of 3722 articles, 728 of which were removed because of duplication. After the initial evaluation, 77 studies were considered eligible after screening the title and abstract. Following the second phase of inclusion assessment, 64 studies were excluded for the sake of conference abstracts (*n* = 8), the inclusion criteria not including advanced maternal age (*n* = 24) or reviews (*n* = 32). The data from two reports were derived from the same trial, so that with the later publication date was excluded [23,24]. One study was excluded due to a lack of sufficient data involving pregnancy outcomes, the method of randomization, and the characteristics of patients [25]. One trial was excluded because the allocation was based on the patients’ decision [26]. Moreover, one trial was conducted to compare the outcomes of embryo biopsy on Day 3 and Day 5 in different age groups, so it was excluded due to the absence of a control group [27]. Finally, nine studies were included in the quantitative synthesis (Figure 1).

### 3.2. Quality Assessment of Included Studies

Among the nine RCT studies included in this meta-analysis, seven trials were considered at low risk of bias on random sequence generation for description of adequate methods for sequence generation (Appendix A). Four trials were deemed at low risk of bias on allocation concealment. In five studies, participants and personnel were not blinded, and one study did not mention the blind status of participants; these studies were considered at high risk and unclear risk of performance bias, respectively. All studies were assessed at low risk of detection bias and reporting bias. Three studies were considered at high risk of attrition bias for lacking mention of intention-to-treat analysis regarding dropouts. One study was at high risk of other bias for the different time windows of embryo transfer in the PGT group and control group.

### 3.3. Description of Included Studies

The main characteristics and outlines of the nine randomized controlled trials included in this study are described in Table 1. In total, there were 2113 women undergoing IVF with or without PGT-A recruited for these trials, and the PGT-A strategies were different in techniques of genetic analysis and stage of embryo biopsy (Table 2). FISH, the most widely used method, was applied in six trials, and CCS was used in the other three trials, including aCGH and NGS. For embryo biopsy, seven trials were performed at cleavage stage, one was at blastocyst stage, and one was on oocytes with polar bodies. Moreover, the definition of AMA also varied in different studies, generally ranging from 35 to 44 years of age. In two trials, the inclusion criteria not only referred to women of AMA but included women <35 years of age with IVF/ICSI attempts or <40 years with repetitive implantation failure [28,29]. In this study, only the data relating to women of AMA were included in the analysis. 

### 3.4. Live Birth

Regarding the different techniques of genetic analysis, three trials (270 events) used with CCS showed that the live birth rate was significantly higher in the women randomized to IVF/ICSI with PGT-A than in those randomized to IVF/ICSI without PGT-A (RR = 1.30, 95% CI 1.03–1.65). However, six trials (275 events) performed with FISH showed a non-significant negative effect (RR = 0.83, 95% CI 0.55–1.25) with a substantial statistical heterogeneity (I2 = 68%), and the pooled analysis of all nine trials (545 events) also showed no difference (RR = 1.01, 95% CI 0.75–1.35) (Figure 2). This suggested that the live birth rate in women of AMA was significantly improved with the application PGT-A with CCS.

Regarding the different stages of embryo biopsy, only the subgroup of blastocyst-stage biopsy showed a higher live birth rate in women with PGT-A than in those without PGT-A (116 events, RR = 1.36, 95% CI 1.04–1.79) (Figure 3), which was a single trial without the result of statistical heterogeneity. This suggested that if the live birth rate in women without PGT-A was 37%, the rate of PGT-A with blastocyst-stage biopsy would be between 38% and 66%. However, no differences were observed in the subgroup of polar body biopsy and cleavage-stage stage biopsy, as well as in all nine trials.

### 3.5. Ongoing Pregnancy

All trials reported the outcomes of ongoing pregnancy. PGT-A with aCGH or NGS showed a higher rate of ongoing pregnancies than the control group (270 events, RR = 1.30, 95% CI 1.03–1.65). Nevertheless, the trials performed with FISH showed a non-significant lower rate of ongoing pregnancies (286 events, RR = 0.83, 95% CI 0.57–1.20). Meanwhile, when trials of PGT-A with FISH were pooled with the CCS subgroup, no difference was proven (RR = 1.00, 95% CI 0.76–1.33). Similar to the result of live birth rate, blastocyst-stage biopsy showed a higher ongoing pregnancy rate in the PGT-A group (116 events, RR = 1.36, 95% CI 1.04–1.79) (Appendix A), while the polar body biopsy subgroup and cleavage-stage biopsy subgroup showed no difference between women with PGT-A and those without PGT-A.

### 3.6. Clinical Pregnancy

Regardless of pooled analysis or subgroup analysis, the clinical pregnancy rate did not differ between women of AMA with PGT-A and the control group in the nine trials (695 events, RR = 0.90, 95% CI 0.71–1.13) (Appendix A). 

### 3.7. Miscarriage

The miscarriage rate was reported in all nine trials. The subgroup of PGT-A with CCS showed a non-significant lower rate of miscarriage (84 events, RR = 0.43, 95% CI 0.18–1.05). Regarding different stages of embryo biopsy, only one trial performed with polar body biopsy showed a significantly beneficial effect in the PGT-A group (41 events, RR = 0.48, 95% CI 0.26–0.89), while cleavage-stage biopsy and blastocyst biopsy, as well as the pooled analysis with all nine trials, showed no significant differences (216 events) (RR = 0.72, 95% CI 0.50–1.03) (Appendix A).

### 3.8. Multiple Pregnancy

Six trials reported the multiple pregnancy outcomes. The pooled analysis showed a non-significant lower rate in women with PGT-A (87 events, RR = 0.72, 95% CI 0.42–1.23) (Appendix A). There was no significant change in multiple pregnancy rate between different techniques and different stages of embryo biopsy.

## 4. Discussion

### 4.1. Summary of Results

The results of our systematic review and meta-analysis could be summarized as two key findings. First, the utilization of CCS in the PGT-A procedure could improve the pregnancy outcomes as it accurately assessed the embryo euploidy status. Second, the blastocyst biopsy might be advantageous in the PGT-A procedure in women of AMA. Although the overall analysis showed that the pregnancy outcomes of the PGT-A group were not better than those of the control group, it was found that the live birth rate, the primary outcome, was higher after IVF with PGT-A than in the control group. Meanwhile, the ongoing pregnancy rate showed the same trend, which was higher in the PGT-A group with CCS compared with those with FISH. Regarding stages of embryo biopsy, seven of the included trials in this study were cleavage-stage biopsy, one trial was polar body biopsy, and one was blastocyst biopsy. In the group of blastocyst biopsy, IVF with PGT-A showed a higher rate of ongoing pregnancy and live birth than that in the control group of women of AMA, whereas comparable rates were observed in the polar body biopsy group and the cleavage-stage biopsy group. 

### 4.2. Comprehensive Chromosome Screening

Compared with FISH, aCGH could detect copy number variations (CNV) and unbalanced translocations effectively by mixing the fluorochrome-labeled test DNA with a control sample and hybridizing them onto an array platform [30]. Fragouli et al. applied both aCGH and FISH to 12 embryos donated from 5 patients [31]. It was observed that the results of nine embryos (75%) were consistent, while two aneuploid embryos were not identified by FISH and were theoretically detected by the probes, and one embryo was recognized with a trisomy of chromosome 8, which was out of the scope of FISH [31]. Additionally, NGS was used in PGT-A as a reliable and high-throughput strategy, which enabled higher sensitivity in the diagnosis of mosaicism with greater resolution compared to aCGH. Various studies validating the accuracy of the NGS approach for CCS of embryos demonstrated a 100% diagnosis consistency with aCGH [21,32]. A comparison between NGS and aCGH applied to PGT-A has also been performed and evaluated [33]. The implementation of NGS for PGT-A revealed higher implantation rates and live birth rates compared to aCGH, which might be attributed to the advantages of NGS in detecting small chromosomal deletions and duplications and mosaicism. In addition, the NGS could be used for the diagnosis of single-gene disorders, translocations, and haplotype analysis in PGT. 

### 4.3. Stage of Embryo Biopsy

Embryo biopsy, obtaining genetic materials from oocytes or embryos, is a significant step during the PGT procedure. It can be performed at different stages, including polar body biopsy (polar bodies), cleavage-stage biopsy (a single blastomere), and blastocyst biopsy (5 to 10 trophectoderm cells). The first and second polar bodies were produced during meiosis of oocytes and seemed to be not necessary for embryo development; therefore, polar body biopsy was considered less damaging than cleavage-stage biopsy and blastocyst biopsy. However, polar body biopsy cannot analyze the genetic information from paternity or the later development stage of embryos, which are important factors affecting its predictive power [34]. More frequently, embryo biopsy was performed on Day 3 by extracting 1–2 blastomeres. Nevertheless, the major drawback of cleavage-stage biopsy was misdiagnosis or missed diagnosis of mosaicism because of the limited materials [35]. With the technical innovation of embryo culture and vitrification, blastocyst biopsy emerged and was conducted by removing 5–10 trophectoderm cells on Day 5, which provided more testing samples than cleavage-stage biopsy for detecting mosaicism and reducing the risk of amplification failure. Moreover, it was reported that the aneuploidy rate was lower in blastocyst biopsy than in cleavage-stage embryos, as euploid cells showed a growth advantage in the embryo development [31]. However, undesirable effects of the embryo biopsy reducing the embryonic development potential have been reported, such as cleavage arrest in polar body biopsy and blastulation delay in blastomere biopsy [8,14,36]. 

### 4.4. Embryo Mosaicism

Embryo mosaicism, a phenomenon of both euploid and aneuploid cells observed in the same embryo, was another significant factor accounting for the ineffectiveness of PGT-A. It was derived from mitotic errors at all post-zygotic stages of the embryo, which increased with maternal age in mitotic aneuploid mosaicism [37]. However, the criteria of mosaicism that the threshold percentage of abnormal cells had been still undefined. Two aspects playing vital roles in the diagnosis of mosaicism were the techniques of genetic analysis and materials obtained from embryo biopsy. Compared to aCGH, NGS was presented with a higher resolution, which could detect mosaicism as low as 20% in aneuploid cells [38]. It was estimated with an incidence ranging from 2% to 13% through the strategy of trophectoderm biopsy combined with NGS analysis [39]. However, their developmental potential remains to be determined. Several studies demonstrated that embryos diagnosed as mosaic were more likely to miscarriage than euploid embryos [39]. Nevertheless, a comparable rate of live birth and ongoing pregnancy was reported between euploid embryos and low-percent mosaicism (<50%) [40]. Accordingly, a standardized assessment is necessary to be finalized for the clinical decision and care of mosaicism in PGT-A.

### 4.5. Study Strengths and Limitations

This study represented a comprehensive synthesis of data regarding the performance of PGT-A in women over 35 years of age. This up-to-date review gave insights into the technical developments of PGT-A procedures, and the meta-analysis provided more reliable summaries with subgroup analysis according to different strategies of PGT-A. Some limitations of this study are worth noting. First, in the included nine studies, only one study involved the determination of mosaic embryos, and it was calculated in all patients 25–40 years of age (rate of 16.8%), lacking data in the AMA subgroup [29]. Hence, the influence of mosaicism on pregnancy outcomes was not analyzed in this study. Second, considering the fact that there was only one trial in the blastocyst biopsy subgroup, and it was tested with NGS, the higher rate of live birth and ongoing pregnancy probably reflected the effectiveness of NGS in PGT-A [30]. Consequently, the effect of blastocyst biopsy requires further study to be verified.

## 5. Conclusions

Contrary to the outcomes of PGT-A with FISH reported in previous RCTs and meta-analyses, the utilization of comprehensive chromosome screening techniques such as aGCH and NGS suggested the beneficial effect of PGT-A in women of AMA. When it comes to strategies of biopsy, blastocyst biopsy seemed to be associated with a better outcome than the polar body biopsy group and cleavage-stage biopsy group. In view of the limited reports, further studies are warranted to confirm the effect of different stages of embryo biopsy on pregnancy outcomes.

## Figures and Tables

**Figure 1 jcm-10-03895-f001:**
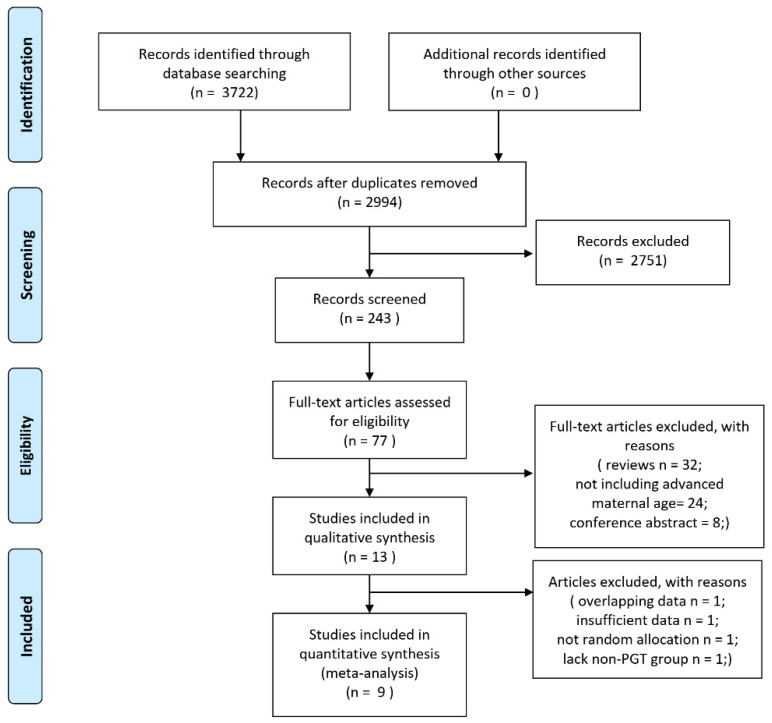
PRISMA flow chart of selection process for meta-analysis.

**Figure 2 jcm-10-03895-f002:**
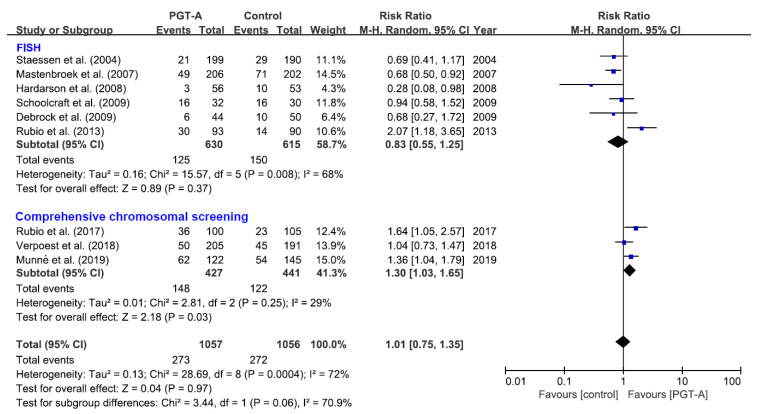
The effect of PGT-A with different techniques of genetic testing on live birth rate. Each study represented by a line. The square represents the point estimate of the effect for a single study, and its area is proportional to the weight of the study. The diamond represents the pooled effect estimate and the width of the dia-mond represents the 95% CI around this estimate.

**Figure 3 jcm-10-03895-f003:**
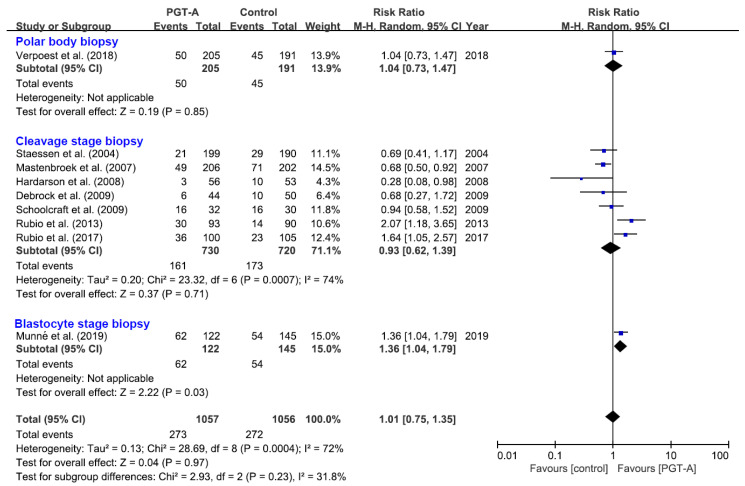
The effect of PGT-A with different stages of embryo biopsy on live birth rate. Each study represented by a line. The square represents the point estimate of the effect for a single study, and its area is proportional to the weight of the study. The diamond represents the pooled effect estimate and the width of the diamond represents the 95% CI around this estimate.

**Table 1 jcm-10-03895-t001:** Characteristics of included studies involving PGT-A in women of AMA.

Study	Patients	Cycles	Inclusion Criteria	Genetic Testing	Embryo Biopsy	Ovarian Stimulation Protocol
Staessen et al., 2004	389	389	≥37 years, both partners with normal karyotype	FISH (chromosomes X, Y, 13, 16, 18, 21, 22)	Cleavage stage biopsy	GnRH antagonist protocol
Mastenbroek et al., 2007	408	836	35–41 years, no previous failed IVF	FISH (chromosomes 13, 18, 21, X, Y)	Cleavage stage biopsy	GnRH antagonist protocol
Hardarson et al., 2008	109	109	≥38 years, at least two or three embryos of good morphological quality if SET or DET, respectively	FISH (chromosomes X, Y, 13, 16, 18, 21, 22)	Cleavage stage biopsy	GnRH antagonist protocol
Schoolcraft et al., 2009	62	62	≥35 years, presence of at least five embryos with ≥6 cells and ≤15% fragmentation on Day 3	FISH (chromosomes X, Y, 13, 15, 16, 17, 18, 21, and 22)	Cleavage stage biopsy	N/A
Debrock et al., 2009	94	94	≥35 years with at least two fertilized oocytes available on Day 1 after oocyte retrieval, and with at least two embryos consisting of six or more cells at Day 3 after oocyte retrieval	FISH (chromosomes 13, 16, 18, 21, 22, X, Y)	Cleavage stage biopsy	Long or short protocols
Rubio et al., 2013	91 + 183	346	RIF in couples <40 years of age with three or more previous IVF/ICSI attempts and transfer of good-quality embryos; 2. AMA in women aged between 41 and 44.	FISH (chromosomes 13, 15, 16, 17, 18, 21, 22, X, Y)	Cleavage stage biopsy	N/A
Rubio et al., 2017	205	205	38–41 years, normal karyotypes, BMI < 30 kg/m^2^, had five or more metaphase II (MII) oocytes obtained from one or two cycles, and had sperm concentrations ≥ 2 × 10^6^/mL	aCGH (fragment larger than 10 Mb)	Cleavage stage biopsy	GnRH antagonist protocol
Verpoest et al., 2018	396	396	36–40 years, BMI between 18–30 kg/m^2^, accepted the transfer of up to two embryos, absence of any type of hereditary condition	aCGH (CCS)	Polar body biopsy	N/A
Munné et al., 2019	661	661	female age 25–40 years undergoing IVF with autologous oocytes with at least two blastocysts of sufficient quality for biopsy and vitrification by Day 6	NGS (CCS)	Blastocyst stage biopsy	N/A

FISH, fluorescence in situ hybridization; aCGH, array comparative genomic hybridization; NGS, next-generation sequencing; CCS, comprehensive chromosomal screening; RIF, repeated implantation failure; AMA, advanced maternal age; BMI, body mass index; SET, single-embryo transfer; DET, double-embryo transfer; N/A, not available.

**Table 2 jcm-10-03895-t002:** Pregnancy outcomes of included studies.

Study	Patients	Live Births	Ongoing Pregnancies	Clinical Pregnancies	Number of Positive Pregnancy Tests	Number of Embryos Transfers	Miscarriages	Multiple Pregnancy	Transferred Embryos	Embryos Per Transfer
	PGT-A	Con	PGT-A	Con	PGT-A	Con	PGT-A	Con	PGT-A	Con	PGT-A	Con	PGT-A	Con	PGT-A	Con	PGT-A	Con	PGT-A	Con
Staessen et al., 2004	199	190	21	29	22	29	22	30	29	39	81	121	7	10	4	6	164	338	2	2.8
Mastenbroek et al., 2007	206	202	49	71	52	74	61	88	81	106	367	364	37	36	10	14	686	756	1.8	1.9
Hardarson et al., 2008	56	53	3	10	3	10	5	13	10	16	45	53	7	6	0	2	70	95	1.5	1.8
Schoolcraft et al., 2009	32	30	16	16	16	16	21	23	25	30	31	30	5	7			68	81	2.2	2.7
Debrock et al., 2009	44	50	6	10	6	10	8	13	11	16	47	49	2	5	1	1	76	88		
Rubio et al., 2013	93	90	30	14	31	17	36	18	40	20	70	74	6	4			152	144	1.6	2
Rubio et al., 2017	100	105	36	23	36	23	37	41	47	48	68	105	1	16	8	3	89	174	1.3	1.8
Verpoest et al., 2018	205	191	50	45	50	45	64	72	72	87	177	249	14	27	14	24	249	440	1.4	1.8
Munné et al., 2019	122	145	62	54	62	54	73	70	88	86			10	16

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
