# Peer review of "Different Strategies of Preimplantation Genetic Testing for Aneuploidies in Women of Advanced Maternal Age: A Systematic Review and Meta-Analysis"

_jcm, 2021, doi:10.3390/jcm10173895_

Round 1

Reviewer 1 Report

The authors did a review report and a meta-analysis of studies using different techniques for chromosomal preimplantation genetic investigations (PGT-A). Following an extensive literature review, nine studies that met the methodological criteria adopted for this review and the meta-analysis were included in the present work. One of the main criteria was a randomized study strategy. They considered within the studies only women of the AMA. They assessed distinct endpoints in groups such as live birth, continuous pregnancy rate, clinical pregnancy rate, multiple pregnancy rate, and miscarriage rate.

Six papers used the FISH methodology and were published from 2004 to 2013. All of these six studies were directed to the detection of a limited set of chromosomes. The paper of Rubio et al. 2017 screened at the cleavage-stage utilizing aCGH. The study of Munné et al. 2019 used NGS for comprehensive screening after blastocyst stage biopsy and lastly the paper of Verpoest et al. 2018 screened polar bodies by aCGH.

One of the major critical points of this paper is the inclusion of 6 studies using FISH for screening. It is a known fact that screening for a subset of chromosomes by FISH is inadequate due to the fact that virtually every chromosome may be affected by aneuploidy at an early stage of the embryonic developmental. It means that a relevant number of chromosomal aneuploidies not detect. The ineffectiveness of FISH analysis as PGT-A has previously been proven, as the authors also stated. The choice to include these studies in this meta-analysis is hardly to follow and in my personal opinion of partial relevance.

For the remaining three CCS-studies only one show a higher live birth rate using an NGS technology. The reasons for this higher pregnancy rate (technology and/or investigated cells) remain unknown and its statistical significance is not assessable due to the fact that the considered work uses a specific NGS technology at the blastocyst stage; in contrast, the other two CCS-studies used aCGH investigating different structures/cells (polar bodies and cleavage stage). In summary, if FISH-based articles are excluded from this review, are the three remaining CCS-studies scarcely comparable to each other and probably insufficient for a meta-analysis.

Finally, reading the article is sometimes not as easy to follow and a style review could increase the quality of the article.

Author Response

Thank you for your advice. We have used Elsevier Language Editing services to improve the English grammar, spelling and diction in this manuscript.

Please find responses in attached file.

Reviewer 2 Report

Overview:

This is a systematic review and meta-analysis with an objective of comparing the pregnancy outcomes of women of AMA after IVF with or without PGT-A

General comments:

Over all a good systematic review and meta-analysis. Clinically relevant. I have some comments suggesting some edits

Other specific comments:

  1. Abstract page 2
    1. In the Participants/material, setting, methods, the ovarian stimulation protocol should be stated in this section.
  2. Introduction
    1. Please state the objective of the study clearly

  1. Material and Methods
    1. Mention more on the definition of advanced maternal age used for this study.
    2. Please include quality assessment – Risk of bias assessment

  1. Results
    1. Table 1: Rubio et al – Have you included data for RIF <40 years? This was one of your exclusion criteria.
  2. Discussion
    1. Discussion section needs reformatting
    2. First paragraph not needed as it is just a repetition of introduction, which can be modified by incorporating some from this paragraph.
    3. First paragraph under discussion should say the key findings from this meta-analysis and how this differs from previous meta-analysis ie state novelty of this study.
    4. State strengths and weaknesses of the study

Author Response

Response to Reviewer 2 Comments

Point 1: Abstract page 2

In the Participants/material, setting, methods, the ovarian stimulation protocol should be stated in this section.

Response 1: Thank you for your suggestion. For the ovarian stimulation protocol, we have added the related information in Table 1 and completed the abstract with the inclusion and exclusion criteria of studies in the method section. Please see lines 21-24.

Point 2: Introduction

Please state the objective of the study clearly

 Response 2: Thank you for your suggestion. The aim of this meta-analysis was to evaluate whether the PGT-A procedure is beneficial for pregnancy outcomes of women of AMA with IVF. We have revised the introduction section to clarify it. Please see lines 69-71.

Point 3: Material and Methods

1.Mention more on the definition of advanced maternal age used for this study.

2.Please include quality assessment – Risk of bias assessment

Response 3: Thank you for your advice.

  1. According to the most current studies (PMID: 32800274), the advanced maternal age is defined as women older than 35 years of age. We have added this definition into the method section. Please see line 83.
  2. Additionally, we have evaluated the quality of included studies using the Cochrane Collaboration’s tool and show the risk of bias summary in Supplemental figure 1.

Point 4: Results

Table 1: Rubio et al – Have you included data for RIF <40 years? This was one of your exclusion criteria.

Response 4: Thank you for your question. In the study of Rubio et al. 2013, the randomized trial was performed in two groups: RIF in couples < 40 years of age (91 patients were enrolled) and AMA in women between 41 and 44 (183 patients were enrolled). We only extracted the data from 183 women of AMA as the statement of study selection in the method. Please see lines 83-84.

Point 5: Discussion

    1. Discussion section needs reformatting
    2. First paragraph not needed as it is just a repetition of introduction, which can be modified by incorporating some from this paragraph.
    3. First paragraph under discussion should say the key findings from this meta-analysis and how this differs from previous meta-analysis ie state novelty of this study.
    4. State strengths and weaknesses of the study

Response 5: Thank you for your advice.

  1. We reformatted the discussion section with subheadings, including summary of results, comprehensive chromosome screening, stage of embryo biopsy, embryo mosaicism and study strengths and limitations.
  2. We revised the first paragraph according to the suggestion.
  3. We stated the key findings of this meta-analysis in the first paragraph.
  4. Strengths and limitations of this study were summarized in the last paragraph.

Round 2

Reviewer 2 Report

The changes are made following the suggestions. thank you.